# Don't Wait, Innovate! Preparing Students and Lecturers in Higher Education for the Future Labor Market

**Marlies Ter Beek** [1,*] , **Iwan Wopereis** [2] **and Kim Schildkamp** [3]

1   Center for Information Technology, University of Groningen, 9747 AJ Groningen, The Netherlands
2   Faculty of Educational Sciences, Open Universiteit, P.O. Box 2960, 6401 DL Heerlen, The Netherlands
3   Faculty of Behavioural, Management and Social Sciences, University of Twente,
    7500 AE Enschede, The Netherlands
*   Correspondence: m.ter.beek@rug.nl

**Abstract:** Technological innovations are changing our society at a rapid pace. The expansion of new technologies (e.g., tools and programs) will inevitably change future jobs in the area of, for example, engineering, healthcare, and science. People working in these areas need digital human capital, which is often acquired through education prior to starting a job. As a result, higher education systems around the globe face increasing demands to prepare their students for the changing labor market. To meet these demands, it is essential to focus on both lecturers' and students' digital competencies. Teaching professionals will have to learn to do new things using new resources. This goes beyond merely replacing work forms and resources; it is a complex process that demands a deeper way of learning in which routines and underlying knowledge and beliefs are explicitly reconsidered. Attention needs to be paid to how lecturers can gradually and continuously develop their professional competencies in the field of educational innovation with IT, to ensure these practices become embedded in future higher education. In this reflection paper, we will discuss key digital competencies for both students and lecturers. We will also focus on how lecturers develop these competencies through effective professional development (PD) activities. Based on a literature review, we present a model for effective lecturer PD with 29 'building blocks'. This model will be used to clarify practical examples of effective lecturer PD aimed at using innovative technology in higher education.

**Keywords:** professional development; higher education; educational innovation with IT; emerging technologies

## 1. Introduction

Technology is an important constituent of teaching and learning practices in engineering academia. Lecturers (as stated by Schildkamp et al. [1], the term "teacher professional development" is mainly used in primary and secondary education. In the higher education context, the terms "instructor" and "lecturer professional development" are most commonly used. In this study, we use the term "lecturer" to refer to teaching professionals in higher education, with the exception of references to existing frameworks and studies) teach about technology while using it. The same holds for students and their learning. Within this academic context, new technologies are often mentioned as an agent and driver of educational innovation [2–4]. They are considered promising tools for solving persistent educational problems, such as demotivation, inflexibility, and dropout, and offer openings for creating an educational system that is increasingly more effective, efficient, and enjoyable for all stakeholders. Unfortunately, evidence-informed techno-optimism is not enough to make innovations successful. A major barrier on the road to successfully implementing technological innovations is the conservative nature of education [5]. Such conservatism probably explains at least partly why innovative strategies through small steps (a so-called

substitutional model) are probably more successful than those that aim at breakthroughs (a transformational model). If the latter is necessary because society demands it, this barrier must be eliminated by taking rigorous measures, such as large-scale policies and incentives that require sustained effort [6–8]. Nevertheless, these transformations can start small. Using a bottom-up approach, focusing on lecturers and their educational context, institutions can foster their digital human capital through effective professional development (PD) trajectories.

However, we do need to acknowledge here that implementing educational technology innovations—especially those not created by its users—is a complex, multilayered, and social process. Just stressing the importance of digital human capital is not enough and will not lead to intensive participation in PD trajectories. This requires changes across several dimensions, including lecturers' values, motivations, and actual practices. For example, how well an innovation fits the values and cultures of lecturers is a more important determinant for change than perceiving a greater benefit [9,10]. The role of the lecturer needs to be placed central as this classic quote demonstrates: "Change is what teachers do and think. It's as simple and as complex as that" [11] (p. 107).

In this reflection paper, we will elaborate on the relationship between technology and the practical use of educational innovations in an increasingly digitized world. Reflections are grounded in (a) the PD literature and (b) experiences with a nationwide project on digitalization in higher education [12]. The purpose of this study is to identify building blocks for effective PD in the use of educational innovation with IT and to showcase how these building blocks can be used in PD programs.

We first discuss educational innovation with media that, according to Bates [13], refers to text, video, computing, social media, and emerging technologies such as augmented/virtual reality (AR/VR), serious games, and artificial intelligence (AI). In this listing, computing is a synonym for information technology (IT). These media find their way into the higher education system and teaching practices. We discuss the proliferation of new media in higher education and refer to issues that also apply to engineering education. We will further discuss the crucial role that lecturers play in the use of new media and the importance of lecturers' PD with respect to educational innovation with media.

The realm of technology as a medium or media is just one side of the 'technology coin', or—as Kirschner and Kester [14] state—one of two cities (cf. *A tale of two cities*, by Charles Dickens). Within this realm, technology refers to technological artifacts or objects. The other side or city refers to technology as design and includes the craft and science of instructional design. These two are precipitated in theories and models such as the ADDIE approach (Analysis, Design, Development, Implementation, and Evaluation; [15]) and the 4C/ID-model (Four-Component Instructional Design-model; [16]). Kirschner and Kester emphasize that a synthesis of the two cities is necessary, which will ultimately result in an engineering science of educational technology [14]. We touch on this synthesis in the second part of this paper. There, we first elaborate on the constituents for the design of PD interventions, which are called building blocks, for effective lecturer development for innovative IT use [17]. Subsequently, we describe a specific type of intervention, called the Field Lab, which aims at implementing (new) IT in teaching practices with the use of a specific combination of these building blocks. We take a closer look at two examples that, respectively, address the use of learning analytics [18] and artificial intelligence [19] by lecturers and students. The latter example will be explored more fully. Lastly, we provide recommendations for research and practice based on our field lab experiences.

### 1.1. Innovation and Digital Competence

An important goal for higher education institutions is to deliver courses that are of interest to the industry and prepare students for the future labor market. This certainly applies to the field of engineering. However, students often experience a 'gap' between the content they learn at their university (or university of applied sciences) and the knowledge and skills that industries demand; in some cases, we cannot even foresee what an industry

will demand in a few decades [7]. Lecturers need to support students in developing the (digital) competencies necessary to improve students' future employability, for example, by using digital technologies or environments to allow learners to develop industry-specific and employability-related skills [20].

Studies on digital citizenship and digital competencies emphasize the role of higher education lecturers being digital citizens themselves [21]. There are several frameworks to describe what it means for lecturers to be digitally competent, such as DigCompEdu [22], the Jisc Teacher profile [23], or the Digital Teaching Professional Framework [20]. All the frameworks include dimensions with regard to (1) designing, executing and evaluating education; (2) equipping students for the digital society of the future; (3) acting professionally as a lecturer; (4) lecturers' digital literacy [24].

Having digitally competent lecturers is crucial, as the process of digitalization that takes place in all spheres of society has a great impact on how we function, both in our private lives and in our professional work. It is no coincidence that this process is often characterized as a digital revolution or digital turn [5,25,26]. The notion that in a computer-dominated world its inhabitants must be able to cope with digitalization, is widely accepted. The tenor is that we need to prepare everyone for digital citizenship, however fuzzy this concept might be [21]. Digital citizenship means being able to participate in full in an increasingly digitized society that is increasingly complex and dynamic. It includes the ability to use digital products and services, and taking advantage of the opportunities they offer, but also anticipating associated change.

Not adequately anticipating or responding to change can have negative consequences. For example, neglecting change can lead to a digital divide, a partition between those who have access to and the use of digital media (or IT) and those who have not, which can also be manifest on an individual or group level [27]. Not adhering to the use of certain generic smartphone applications such as WhatsApp, WeChat, Line, or Telegram, for instance, can lead to alienation within (professional) groups. The implementation of new ITs can also result in technostress, which can be defined as "an adaptation problem caused by individuals' incapability to cope with new ICT and requirements associated with the use of ICT in a healthy way" [28]. Techno-overload, techno-complexity, techno-insecurity, and techno-uncertainty are found to be enablers of such stress. Factors that inhibit technostress include advancing digital literacy, providing technical support, and promoting involvement. In order to be successful, policy related to these inhibitors should be sustained and institutionalized. Although technostress is a phenomenon that is not new, the recent outbreak of the COVID-19 pandemic made clear that the aforementioned inhibitors are even more relevant in times of rapid change. The sudden shift to online learning in higher education, for example, resulted in emergency remote teaching (ERT) programs that (fully understandably) lacked didactical rigor [29]. Despite all the good intentions and first aid programs (e.g., [1]), this sudden transformation led to much dissatisfaction and uncertainty among lecturers [30]. It also confirmed that adequate support in the case of technological change is indispensable.

ERT is meaningfully different from the goals of well-thought-out blended learning designs aimed at preparing students for future jobs [29]. The use of innovative technology in education implies new teaching and learning practices, and, therefore, requires both organizational and personal changes; lecturers need to develop new knowledge, beliefs and competencies [31]. Research has shown that lecturers' personal backgrounds (i.e., years of work experience), their use of social networking sites, and internet self-efficacy are significant predictors of their digital citizenship [32]. The experiences during the COVID-19 crisis have indicated that many lecturers in higher education experience difficulties regarding the mix between online and face-to-face education [33] or in coping with ERT [1,29]. Designing effective (blended) learning activities is not something you learn overnight; therefore, it is crucial for the quality of future education that institutions facilitate long-term and ongoing professional development with regard to innovative IT use in higher education [34–36]. This provides lecturers with ample time to learn, practice, apply what they have learned,

and reflect on new teaching strategies [17]. This is also a focus of the Dutch "Acceleration Plan Educational Innovation with IT" [12].

### 1.2. The Acceleration Plan

This reflection paper draws on the work of a nationwide educational program in the Netherlands called the "Acceleration Plan Educational Innovation with IT" [12]. This program, commonly referred to as the "Acceleration Plan" (AP), was initiated before the pandemic to support higher education institutions focusing on three main goals: (1) to improve connections with the labor market, (2) to create more flexible education, and (3) to improve and enhance learning with innovative technologies. In total, 39 Dutch higher education institutions, both Universities and Universities of Applied Sciences, worked together on these goals in eight thematic zones and three working groups (i.e., teams consisting of representatives from participating institutions). The participating members included, among others, educational designers, educational technologists, lecturers, professors, and support staff—all of them with a professional and/or personal interest in the use of IT in higher education. The program ran from January 2018 to December 2022.

One of these zones is called "Fostering digital human capital". In collaboration with Dutch education and research institutes on one side and various companies, institutes, and governments representing the labor market on the other side, this zone is concerned with identifying and sharpening crucial digital competencies for students that need to be addressed in their bachelor's or master's programs. The goal is to align both sides on the necessary digital competencies for life and work, enhancing the future-proof flow from student to professional. This approach has been based on the European Digital Competence Framework for Citizens [37]. This framework consists of five competence areas: (1) information and data literacy (e.g., browsing, searching, and filtering data, information, and digital content); (2) communication and collaboration (e.g., interacting through digital technologies); (3) digital content creation (e.g., integrating and re-elaborating digital content); (4) safety (e.g., protecting devices); (5) problem solving (e.g., identifying needs and technological responses).

It also requires certain lecturer competencies to be able to help students to become digitally competent citizens. Lecturers were the focus of another zone of the AP, namely, the zone called "Facilitating professional development for lecturers". The goal of this zone is to improve the quality of education by supporting institutions in enabling all lecturers to make effective use of IT in delivering instruction to their students. The zone, "Facilitating professional development for lecturers" (e.g., "lecturer professional development" or "PD") supports institutions in a process of improvement based on a collection of proven and effective PD strategies. This is because acceleration actually takes place within the institutions.

## 2. Materials and Methods

### 2.1. A Model for Effective Lecturer Professional Development with IT

Educational innovation involving IT has been a priority for higher education institutions for many years [6,38,39]; however, many institutions have been struggling with their digital transformation strategies, particularly with designing effective PD focused on educational innovation using IT [1,40]. Of course, pioneering lecturers exist who are ahead of the game, constantly and proactively improving the content of their own lectures and instruction using the latest technological resources. Nevertheless, it is important that all lecturers continuously work on improving the quality of their teaching; therefore, facilitating PD for lecturers is crucial for the quality of education.

Over the last two decades, there has been an increase in the research literature on lecturer PD (e.g., the study by Darling-Hammond et al. [41]). PD programs can be defined as "systematic efforts to bring about change in the classroom practices of lecturers, in their attitudes and beliefs, and in the learning outcomes of students" [42] (p. 381). Gradually, the research focus shifted towards preparing (preservice) lecturers for IT use [43–45],

online education [46], and blended education [47]; however, literature on the specific effective characteristics of PD in higher education is scarce, and there are few studies that combine all three elements: lecturer PD, educational innovation with IT, and the context of higher education [36]. Therefore, the "Lecturer PD zone" explored which building blocks could be identified in the literature and by experts in the field that helped provide effective professional development for higher education lecturers in relation to educational innovation with IT.

Research on lecturer PD in all educational sectors has grown. Various studies have identified several effective components of lecturer professional development [48]. In addition, there is increasing attention for educational innovation with IT and how lecturers can be supported [35,36,49,50]. Nevertheless, studies on the specific effective features of professionalization in higher education are scarce; therefore, we conducted an explorative literature review to collect studies published between 2010–2020 including a combination of three components: (1) lecturer PD, (2) educational innovation with technology, and (3) higher education [17]. The initial search used combinations of terms such as professional development, technology, and digitalization and was carried out within the Education Resources Information Center (ERIC) database and within the internal documents of institutions participating in the AP. After scanning the abstracts for relevance (i.e., whether they described each of the three components mentioned above), 26 relevant publications were included. Additionally, we specifically searched for reviews in the field of lecturer PD on educational innovation using technology. Based on these 26 articles and additional review studies, initially 89 effective characteristics (or 'building blocks') of lecturer PD were found. These results were presented to six experts in the field of lecturer PD and educational innovation with IT in higher education. The experts suggested the merging of elements in order to create a workable amount of building blocks. The prototype model was used in educational practice and refined based on these practical experiences.

The study resulted in a model (see Figure 1) that provides institutions and educational designers with the tools to help them design and evaluate effective PD activities in the field of educational innovation with IT. The identified building blocks were clustered in three domains based on the main constituents of instructional design theories [51]: (1) lecturer characteristics (e.g., prior knowledge), (2) characteristics of the professional development (e.g., active participation), and (3) organizational characteristics (e.g., vision and policy towards educational innovation with IT) [17].

The first domain, the *lecturer characteristics*, relates to the competencies of lecturers with regard to PD in educational innovation with IT. It is not about the background characteristics of lecturers (such as age or number of years of work experience), but about what lecturers think, what they know, and what they are able to do. It is important to take into account these lecturer characteristics in the PD process to enable tailor-made activities. For example, if the facilitator of the PD program notices that the needs of the participating lecturers are not adequately met, he or she may decide to make certain changes on the spot. The second domain, the *characteristics of professional development*, contains building blocks that are related to the form and didactics used or the content of the PD activity. The third domain, the *characteristics of the institution*, refers to the context in which the PD takes place. This concerns the service and support provided by the educational institution and the national measures that impact educational innovation with IT. Although each building block contributes to effective lecturer PD, they presumably cannot all be applied simultaneously. In an educational design research process, based on a needs analysis, the PD developers can select the building blocks that are most suitable given the lecturer characteristics in a certain context and select the characteristics of the professional development that may be useful given the target of the PD (e.g., blended learning, artificial intelligence, or learning analytics).

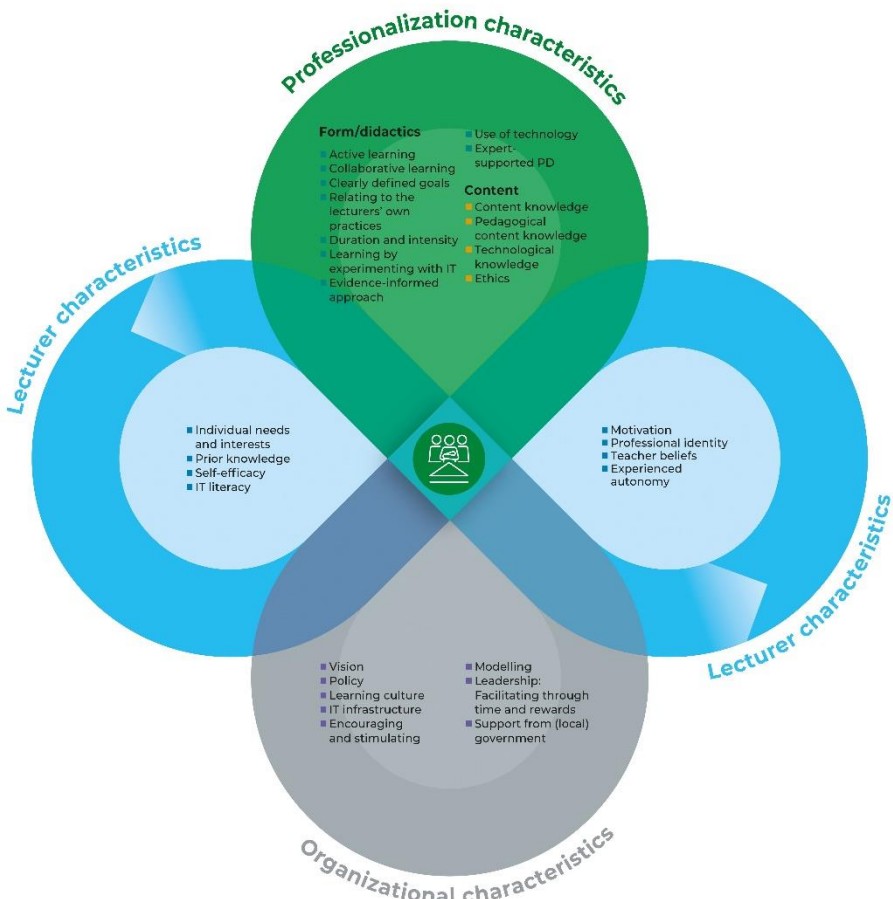

**Figure 1.** Model with building blocks for effective professional development [17].

## 2.2. Effective Lecturer PD in Practice: Field Labs

Based on the model for effective professional development, the "Lecturer PD" zone developed various field labs to stimulate the use of emerging technologies in higher education [52]. They combine good practices that are already known in higher education with effective building blocks from the model shown in Figure 1 into experimental PD settings. The development process of the field labs was based on the Educational Design Research (EDR) method [53]. This is a design-oriented research approach in which interventions are developed and studied in close consultation with practitioners. Central to the method is the iterative development of solutions to practical and complex educational problems, based on scientific inquiry. The process consists of three main stages: (1) an analysis and exploration stage, (2) a design and construction stage, and (3) an evaluation and reflection stage. EDR also seeks to discover new knowledge that can inform the work of others facing similar problems; therefore, field labs provide a source of information for educational designers and PD facilitators in higher education.

Field labs consist of freely downloadable manuals, including guiding materials, to set up a PD trajectory with regard to themes such as blended learning, artificial intelligence, and learning analytics. They come in various forms, including professional learning communities and a hackathon. Educational designers or PD facilitators can act as a facilitator and use the field labs to create their own PD activities and the manuals provide a generic basis for an institution-specific application. Table 1 shows the six developed field labs and their characteristics. The building blocks apply when there is specific attention paid to them within the design of the field lab, for example, by the nature of certain activities or assignments. It shows that some building blocks are present in each of the field labs (e.g.,

Active learning), whereas other building blocks are only explicitly paid attention to in a few field labs (e.g., Ethics).

**Table 1.** Field labs of the Acceleration Plan and the building blocks they focus on primarily.

| Theme | Form | Primary Building Blocks |
|---|---|---|
| AI in higher education | Hackathon (with preparation modules) | Active learning; Clearly defined goals; Collaborative learning; Ethics; Expert-supported PD; Learning by experimenting with IT; Technological knowledge; Use of technology |
| Designing blended education | Professional learning community | Active learning; Clearly defined goals; Collaborative learning; Duration and intensity; Evidence-informed approach; Relating to lecturers' own practice; Technological knowledge |
| Digital peer feedback | Professional learning community | Active learning; Clearly defined goals; Collaborative learning; Evidence-informed approach; Expert-supported PD; Relating to lecturers' own practice; Technological knowledge |
| Formative assessment | General and in-depth sessions based on the formative assessment cycle, led by a facilitator | Active learning; Clearly defined goals; Collaborative learning; Evidence-informed approach; Expert-supported PD; Relating to lecturers' own practice; Technological knowledge; Use of technology |
| Learning analytics | Six sessions covering different themes with regard to LA, led by a facilitator | Active learning; Clearly defined goals; Duration and intensity; Ethics, Relating to lecturers' own practice; Technological knowledge |
| Open Educational Resources (OER) | Six sessions covering different themes with regard to OER, led by a facilitator | Active learning; Collaborative learning; Relating to lecturers' own practice |

Note. This table only shows the building blocks related to the domain of PD characteristics.

In the following paragraphs, we will describe how the building blocks were used in designing two field labs: Learning Analytics and AI in higher education. The field lab, AI in higher education, has already been used in practice by some of the participating institutions, and, therefore, provides us with preliminary results about how lecturers experienced the building blocks within this field lab. These experiences will serve as a tentative first example of how the building blocks can be applied in various higher education contexts.

### 2.2.1. The Field Lab Learning Analytics

Learning analytics (LA) involves the measurement, collection, analysis and reporting of data on students and the context they find themselves in, with the aim being to understand and improve the learning of students and the learning environment. Despite the fact that LA is increasingly available within field labs and other experimental environments, its adoption by lecturers at the class level is not yet commonplace. Yet wider adoption is important; only once LA is operationalized in the authentic teaching environments of our lecturers and students will this lead to real-world improvements. Shibani et al. [54] describe a wide range of factors that contribute to the institutional adoption of LA. Four of these factors are specifically relevant for adoption at the lecturer level. First of all, it is important to involve lecturers early on in the process by inviting them to participate in the design of the LA within the education process (co-design) and in further iterations. Secondly, this design should also lead to authentic learning situations, in which technology is not used artificially but in a way that is relevant to education (authentic experience). Thirdly, it is important that lecturers are encouraged to use learning analytics and that they know they can count on proactive support (empowerment). Finally, it is important that cross-pollination takes place with lecturers sharing their knowledge and experience of learning analytics with their peers (future adoption). Thille and Zimarro [55] endorse this

vision of cross-pollination. They even advocate the development of LA in a collaborative process between different institutions.

In line with the advice by Thille and Zimarro [55], the field lab, Learning analytics, was designed and created by a multidisciplinary team of educational researchers, IT specialists, and lecturers from various higher education institutions in the Netherlands. The field lab has three main learning objectives. After completion, a lecturer will be able to: (1) use LA in a well-substantiated and ethically responsible manner; (2) use LA to provide students with insight into their own learning process so that they can work together to fine-tune this learning process based on the information gained; (3) use LA to reflect on their own teaching practice and make improvements where possible. In six proposed sessions, starting with the basic definition of what LA entails and what data is available, the participants work towards the practical application of LA in their own educational context; working together with lecturers to ensure that the application of LA aligns with their educational practice is considered pivotal to its successful implementation [54].

The first session covers the basics of study data, learning analytics, and the opportunities that they offer for the learning process of students. The second session focuses on the data available within the institution. If you wish to use learning analytics, you have to take account of data protection legislation and the ethical boundaries that apply. To gain a better feeling for this, a group discussion of what is and what is not permitted is a key part of this session. The third session is about the potential pitfalls concerning data analysis and the conclusions that you draw, including issues such as validity, the evidence-based interpretation of study data, and good practices from teaching practice. The fourth session focuses on using student data to benefit students' ability to self-regulate. During the fifth session, lecturers will take a critical look at their course data and will formulate an improvement plan. Finally, during the sixth session, the information from the previous sessions comes together while the participants discuss how LA can become a part of the students', lecturers', and the institution's culture.

The design of the field lab entails multiple building blocks from the model presented in Figure 1. With regard to form and didactics, there is *active learning* involved: for example, participants engage in a role-playing activity in the final session. Lecturers will *relate* the use of *LA to their own educational practice* based on *clearly defined goals*; in the third session, lecturers will use LA to meet their own goals formulated earlier. Since the second session concerns what you can and cannot do with data, the building block of *ethics* is involved as well. Additionally, lecturers will expand and use their *technological knowledge*. Lastly, the building block *duration and intensity* is involved, because the field lab spans a time period of several months, during which the lecturers are enabled to implement LA in their own curricula. With regard to lecturer characteristics, the field lab takes into account *individual needs and interests*, lecturers' *experienced autonomy, prior knowledge,* and **teacher beliefs** (i.e., opinions and beliefs about what 'good teaching' entails).

### 2.2.2. The Field Lab AI in Higher Education

Artificial intelligence (AI) is making a significant advance in higher education worldwide. Higher education institutions have recorded measurable results when they implement AI [56]. This is why expectations about the role of AI are high, for example, with regard to the workload of lecturers, personalized learning, the effectiveness of digital learning resources and the generation of substantiated understanding of student performance. This field lab is focused on the practical possibilities that AI can offer in higher education, and on its consequences for lecturers and students. The format of this field lab is different from the field lab, Learning analytics, since it is not a long-term trajectory for a professional learning community, but a short event (a hackathon) which uses a pressure-cooker format to engage participants deeply in a short amount of time.

In June 2021, we organized this hackathon, in which four teams from different higher education institutions competed against each other. The teams consisted of 4–5 educational designers, IT specialists, and/or students from four different HE institutions. The

supervised part spanned two half-days, but the teams also worked independently during the evenings. The field lab consisted of two parts: an optional online preparation module and a live event (the hackathon). The participants prepared themselves by studying the resources contained in the preparation module before they moved on to solve a practical case study concerning AI in higher education. In this way, they learned the basics of AI and were challenged to do so in the context of their own working environment. During the hackathon design phase, the participants were able to consult Jedis (i.e., experts in the field of AI) if needed. The teams built proofs-of-concept and presented these on the last day of the hackathon. Their concepts were judged by experts in the field. The winning concept incorporated AI to link apps that measure students' stress indicators.

Compared to the field lab, Learning analytics, the design of this field lab incorporates similar building blocks, such as *active learning, clearly defined goals*, and *technological knowledge*; however, the hackathon format results in the inclusion of additional building blocks. Since participants collaborate in teams during the hackathon, there is *collaborative learning* involved. The Jedis, who support the teams if needed, are a clear implementation of the building block of *expert-supported PD*. For the hackathon assignment, the participants are requested to take *ethics* into account. Lastly, participants build a proof-of-concept using programs such as Python, TensorFlow, or PyTorch, as is reflected in the building blocks, *learning by experimenting with IT* and *use of technology*.

### 3. Experiences of the Participants

At the moment of writing, several field labs are being conducted at various Dutch higher education institutions as pilot studies. Following the EDR method, it is our aim to evaluate and reflect on the field labs with the participating lecturers and facilitators. To do so, we use the Kirkpatrick model (also known as the four levels of learning evaluation; [57]) to focus on lecturers' reactions (RQ1: How did they experience the field lab?), learning (RQ2: What did they learn?), behavior (RQ3: What did they apply in practice and how?), and results (RQ4: How did it affect student learning outcomes?). The participants and facilitators receive an evaluation questionnaire where they can indicate agreement with multiple items on a 5-point Likert scale (1 = completely disagree, 5 = completely agree). There is also room for elaboration and sharing experiences in an open-ended question.

As stated earlier, the field lab, AI in higher education, is the first field lab used by lecturers of multiple organizations and of which evaluation results are available. A small number of participants ($N = 4$) filled out an evaluation questionnaire after the hackathon. Descriptive analyses showed that the building blocks collaborative learning, expert-supported PD, and learning by experimenting with IT were valued most. According to the participants, useful elements of the hackathon were 'the fact that this format pulls me out of my comfort zone', 'the accessibility of the tutorials', 'a quick way to get to know AI', and 'working together with colleagues from different study programs or academies on one specific assignment'. The main constraints were time and the fact that it was not possible to build the AI application in two days: 'We haven't gotten around to actually building [it]. We didn't have the knowledge, tools or time for that'. One participant suggested to split the hackathon into a concept phase and a programming phase: 'The pace may be too high. Conceptual progress is also helpful, so distinguish a level of concept and programming'; however, the participants were very motivated to get as much work done in this limited amount of time, as one of the participants stated: 'We worked throughout the night to finalize our plans and to develop the wireframes'. The experiences of the participants indicated that the hackathon had been a positive experience and resulted in a (more) positive attitude regarding the possibilities of AI in education. The next question that needs to be answered is whether this form of PD is an effective way to introduce lecturers to innovative technology, such as AI.

Due to restrictions caused by the COVID-19 pandemic, other field labs have been implemented on a small scale or in alternated forms, often deviating from the proposed program. In addition, the response rates for the various questionnaires of the field labs

were very low. The following field labs have been implemented and evaluated: designing blended education (*N* = 6), formative assessment (*N* = 2), digital peer feedback (*N* = 1), and learning analytics (*N* = 0), resulting in a total number of 13 completed questionnaires for all field labs (including the AI hackathon); therefore, it is difficult to draw firm conclusions based on these results. Nevertheless, some preliminary patterns can be identified from the formal and informal data. The evaluations of the initial use of field labs in Dutch higher education institutions (period: January 2020–September 2021) show that time and resources—the building block Leadership: Facilitating through time and rewards—are considered to be crucial when it comes to lecturer PD in the field of innovative technology. PD facilitators, lecturers, and students need sufficient time to master new tools, programs, or applications. Time is also required to make connections between didactics, pedagogy, and technology. Next to that, the building blocks that are valued most by participants are relating to lecturers' own practices, active learning, and collaborative learning. It should be noted that face-to-face meetings, in which participants can collaborate, are especially appreciated. Effective lecturer PD in the context of educational innovation with IT does not only start with the right instruments, but also with physical interaction and communication between the participants.

## 4. Conclusions and Discussion

Both lecturers and students need to be prepared for the future labor market. Various studies and frameworks highlight the importance of digital competencies such as digital literacy and the use of innovative tools; however, the transition from innovative technology use in the industry to existing higher education programs often happens rather slowly. Industries and higher education institutions need to cooperate and communicate extensively to train students adequately for the jobs of the future and lecturers play a key role in this transition. Educating lecturers to apply and integrate innovative technology in their curricula paves the way to future-proof education. It is up to higher education institutions to facilitate effective forms of lecturer professional development in the field of innovative IT to enable lecturers to scale up the pace of change. Despite the technical character of professionalization focused on educational innovation with IT, it seems that facilitation in terms of time, rewards, and a collaborative learning culture is an important precondition for this process to succeed. This is a finding that is similar to the results of a recent study by Hubers et al. [58]. Additionally, a reliable IT infrastructure and the availability of adequate educational and technical support are indispensable for flexible lecturer PD.

The building blocks for effective lecturer PD described in this article, as well as the materials from the field labs, can inspire educators, educational designers, IT or educational support staff, and even board members to reflect on lecturer PD in the context of future-proof learning. In this paper we presented the first experiences of working with the building blocks and field labs. There are many questions that still need to be answered. For example, which of the building blocks are a "need to have" in any PD (independent of the IT topic) and which are "nice to have" in order to foster the IT competence development of lecturers? In our digitalized society, the building block 'ethics' would probably be considered as a "need to have" building block in any field lab. Yet, not all of the field labs designed for the AP paid (explicit) attention to this building block. The question here is, what are the consequences of not using this building block (explicitly)?

Further research could also focus on fidelity. The field labs have been designed in such a way that organizations can adapt the field labs to their own context. This means that most field labs will probably not be implemented with a fidelity perspective in which the field lab is performed exactly as intended [59]. The focus here is a local adaptation perspective, in which adjustments are allowed, as long as organizations adhere to the core components of the field lab [60]; however, the question here is what happens when organizations adapt the field lab to their own context, leaving one or two or even three building blocks out of their re-design? How high is the 'tolerance' of the field labs in order for them to still be effective? Tolerance refers to how precisely the design intentions must be enacted for

the field lab to be true to its goals [53]. Too many adaptations can be counter-productive leading to "lethal mutations" [Brown and Campione, 1996, cited in 53], not resulting in the development of the desired IT competences of lecturers.

Even after a year of fully online education, in which many colleagues have shown themselves to be resistant and able, the PD of lecturers in the use of IT remains of great importance. The model with building blocks and the field labs presented here can be used as a starting point for further research. Moreover, we hope that the reflections offered in this paper can provide an evidence-informed base to initiate substantive conversations, reflect on lecturers' PD, and take the first steps towards building human capital both within and outside the institution. After all, digitally competent lecturers are needed for educating digitally competent students.

**Author Contributions:** Conceptualization, M.T.B., I.W. and K.S.; methodology, M.T.B.; formal analysis, M.T.B.; investigation, M.T.B., I.W. and K.S.; resources, Acceleration Plan; data curation M.T.B.; writing—original draft preparation, M.T.B., I.W. and K.S.; writing—review and editing, M.T.B., I.W. and K.S.; visualization, M.T.B., I.W. and K.S. All authors have read and agreed to the published version of the manuscript.

**Funding:** This research was funded by the Dutch Ministry of Education, Culture and Science.

**Institutional Review Board Statement:** The study was conducted in accordance with the Declaration of Helsinki, and approved by the Ethics Committee of the University of Twente (protocol codes for the different evaluations of the field labs: 201487 and 201867 approved on 21 December 2020 and 11 June 2021.

**Informed Consent Statement:** Informed consent was obtained from all subjects involved in the study.

**Conflicts of Interest:** The authors declare no conflict of interest.

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
