# Peer review of "Don’t Wait, Innovate! Preparing Students and Lecturers in Higher Education for the Future Labor Market"

_education, doi:10.3390/educsci12090620_

Round 1

Reviewer 1 Report

This paper addresses a highly relevant research (and practice) question, namely how to design & implement effective Professional Development for Higher Education lecturers when it comes to the innovative use of ICT in teaching, in order to empower them to contribute to future-proof education for their higher education students. I think that the model developed in this paper is highly relevant and should lead to fruitful applications in many contexts, in addition to the (field lab) use cases presented in this paper. It is however a bit a pity that the authors could only present preliminary results here. But that is what happened to many educational researchers during this global pandemic and I do think that these preliminary results are worth sharing and the authors were cautious (enough) to not draw too quick conclusions from them.

Overall, very well written with a clear argumentation structure and precise vocabulary, and up-to-date references. It was a pleasure for me to read this manuscript.

I just have some suggestions for small changes to share with the authors to improve their manuscript.

11 People working in these areas need digital human capital,

12 which can be acquired through proper education.

Suggestion: these competences can also be developed via means other than "proper education", like self-teaching, peer-learning, etc. So maybe you want to include the diversity of opportunities and ways to develop them or say something along the lines of "among others".

49 In this listing, computing is a synonym for information and 

50 communication technologies (IT)

Replace (IT) with (ICT) ort change to Information Technology, since you're using IT all over the chapter.

359 RQ3: hat did they apply in practice and how? => RQ3: What did they apply in practice and how?

Author Response

Dear reviewer,

Thank you very much for your kind words and suggestions to improve the original manuscript. Below we have added our responses to your questions and comments. The changes in the manuscript have been made accordingly and should be visible in the revised version of the document.

We are indeed cautious not to draw any quick conclusions from our research, since we have found only preliminary results. To emphasize this even more, we have added a few sentences stating the reflective and informative nature of this article, and the fact that the paper and the model in it can contribute to applications in various Higher Education contexts, as you suggest.

The use of ICT has been changed to IT throughout the manuscript to add to its consistency.

Reviewer 2 Report

This paper offers an interesting approach towards digital technology-led innovation in higher education, aiming the debate on the pivotal role of lecturers. It covers an up-to-date theme, drawing on data of an ongoing project in the Netherlands. In general, the paper is well written and dialogues with very recent literature. Also, author(s) reinforce the topic’s importance by linking it with very recent social happenings such as COVID-19. When it comes to the paper’s aims, structuring and organization, some issues deserve special attention in order to bridge a few content and format gaps. I would suggest taking into consideration the following aspects:

- The word “chapter” appears a few times along the manuscript, including in the abstract, raising doubts if this is an original text; it might be the case it is just a language misuse issue – please review it;

- The keyword “student learning” does not seem very suitable for this article, considering the debate it conveys;

- Introduction refers to a conservative nature of education as a barrier to the implementation of technological innovations. Both this section and the following ones fail to discuss the role of “change” itself in processes of innovation. Literature reports, for example, that rather than an issue of competence, digital transformation in education antagonizes with a repeated and constant demand for change which drives teachers/lecturers to turn it down. Some problematization in this direction would help strengthening the argument later built that bottom-up approaches are smarter in addressing the fostering of digital human capital;

- The program Acceleration Plan Educational Innovation with IT should be referenced in its first mentioning, in section 1.1;

- When the Acceleration Plan is described in section 1.2, it is said that 39 Dutch HEIs participate in it, but there is no mention concerning number of participants, only a general profile of who they are. Likewise, when more specific information is provided about the field labs, there are no mentions on specifics of the participants. This makes it hard to grasp some of the preliminary results advanced in the article. It is a delicate issue, especially when the model with building blocks assumes lecturer’s characteristics as one if its core dimensions;

- The discussion on materials and methods mentions a literature review that does not seem to be very systematic. Even if not following the methodological approach of a systematical review, it is important to clarify steps concerning this procedure. It is said that after scanning the abstracts for relevance, 26 publications were included (section 2.1). The presented protocol is too simplistic and a bit inconsistent, though. The used key words are too straight (and that is a fragility of the protocol), there is no information on which databases the search was carried out, as well as the manuscript lacks information about the total amount of articles consulted and the reasons why part of them was eliminated. This information is crucial since it is based on this review that it emerges a framework of “89 effective characteristics (or ‘building blocks’) of lecturer PD”. By the way, we have no further reference about the materials that provide grounds for this framework. A table synthesizing this step would benefit the consistency of the framework;

- Figure 1 lacks quality especially concerning the text part;

- The idea that the building block ‘Ethics’ only occasionally is part of the field labs is somehow worrying; a close look to the model will lead the realization that some blocks will always be part of the dynamics while others might change depending on actions’ focus. Not acknowledging or discussing this fact fragilizes the proposed model;

- At the end of section 2.2, we read that in the coming subsections, it will be explained “how the building blocks were used in designing the field labs Learning Analytics and AI in higher education”, but there is no reasoning for this. Why does the focus rely on these two labs? How are they representative of the ongoing work? What makes them special? The random presentation of these two field labs makes the article a bit inconsistent;

- It is noteworthy the absence of information regarding the participants of field labs. Even if the argument links to a desired focus on the labs as a PD model, the text is incoherent, because the sections that follow discuss preliminary results considering participants’ active involvement in data collection processes, therefore, knowing their profile is essential;

- Lines 290/291 affirm that “participants work towards the practical application of LA in their own educational context, which is supported by the second factor of Shibani et al. [49]”. What is that supposed to mean? This paragraph needs some rewriting;

- It would be important that preliminary results focused more on data already produced rather than presenting how analysis will proceed. More detail on the questionnaire that was applied to field labs participants (how many respondents; who are they, etc.) would be more relevant than a discussion of further steps to be taken – actually, there is a mismatch between this section and the methodological one (much focused on the building of a PD model); this preliminary results section would benefit of an entire restructuring focused on data already produced. It is hard to understand how the author(s) concluded that “this experimental form of professional development can be regarded as a potentially effective way to introduce lecturers to innovative technology” based on the provided data;

- Discussion (which is actually a wrap-up) could also be improved by establishing more specificities of how the analyzed program contribute to the thinking of digital competences fostering in the context of higher education, without losing sight of the focus on lecturers’ PD.

Author Response

Dear reviewer,

Thank you very much for your kind words and suggestions to improve the original manuscript. Below we have added our responses to your questions and comments. The changes in the manuscript have been made accordingly and should be visible in the revised version of the document.

First of all, it is important to explain that our work has not been published before, but was originally intended to be a book chapter. The editors of the book decided that a journal would be more appropriate for the papers collected. That is why the term ‘chapter’ accidentally still appears throughout the manuscript. We have changed these occurrences into terms such as ‘this article’ or ‘this paper’.

Second, our contribution aims to describe the developed model and its first application in Higher Education practice. We realize that the formatting style of the journal (e.g., the use of fixed headings such as ‘Materials and methods’ and ‘Results’) might give the impression that our study aims for results in terms of effectiveness. We are, however, cautious not to draw any quick conclusions from our research, since we have found only preliminary results and these are based on the implementation of only one field lab. To emphasize this even more, we have added a few sentences in the introduction and conclusion sections, stating the reflective and informative nature of this article, and the fact that the paper and the model in it can contribute to applications in various Higher Education contexts. We also highlight that further research is needed and the reflections in this paper can be used as a starting point.

In the same line of thought, we do not claim to have conducted and/or present the results of a systematic literature review. Therefore, we now emphasize in the text that the exploratory literature review functioned as the basis for the building blocks presented in our model. A sentence on the (scientific) databases used for this literature search has been added for clarification. Also, we now refer to the initial publication (which provides more detail on the search procedure) earlier in the text.

We do agree with you that some detailed information about the characteristics of the field lab participants was lacking, given that lecturer characteristics are the core part of our model and that the experiences of these lecturers provide the basis for our preliminary results. However, we have already added most of the information we know about these participants. Unfortunately, we have not collected detailed information about some of the participants’ characteristics (e.g., participants’ prior knowledge), since the data were collected for formative evaluation purposes so that we could improve the quality of our field labs and gain more insights into the building blocks in action. Moreover, we did not want to burden lecturers with extensive data collection during the COVID-19 pandemic, a period in which lecturers’ workload was already very high.

In our conclusion and discussion section, we added two paragraphs discussing further research which would be needed with regard to digital competences fostering of lecturers based on our model and field labs.

Other changes to the manuscript are:

  • The keyword “student learning” has been removed and replaced by “higher education”. Additionally, we changed the keyword “professionalization” to “professional development”.
  • As you suggested, we have added a section in the introduction on the role of change and how difficult implementation of (PD in) educational innovation with IT is, with a focus on the role of the lecturer.
  • We have added information about the characteristics of participating members of the Acceleration Plan in general, as well as the numbers of evaluation questionnaires that have been filled out for each field lab.
  • We have added an explanation for the selection and description of the results and experiences of two field labs in section 2.2.
  • The paragraph on using learning analytics in practice, including the reference to Shibani et al., was rewritten.
  • The application of building blocks within the field labs (Table 1) is explained more clearly – indeed, ethics are incorporated in each field lab, but in some it is more explicitly referred to (i.e., it is one of the central themes of the sessions) than in others. This is now acknowledged, as you suggested.
  • According to your comment, the discussion section has been supplemented with more specificities of how the analyzed field labs contributes to fostering digital competences in the context of higher education and lecturers’ PD.

Round 2

Reviewer 2 Report

The amended version of the article responds to most of the concerns I addressed before, and I am very pleased with the new reading of the manuscript. The introduced changes provided more consistency to the paper, which now assumes its substance – it is indeed a reflection paper – and establishes more clearly its goal. Although some reservations arise as we now know the number of participants – and responses - of the field labs, the fact that the study states it with transparency and assumes that “it is difficult to draw firm conclusions based on these results” (lines 420/421), combined with its positioning as a reflection paper, guarantees its liability. Moreover, the new reflections included in the conclusion section ensures the paper relevance concerning its role as an instrument to guide reflection concerning “lecturer PD in the context of future-proof learning” (lines 453/454). I would only reinforce the need to improve quality on the graphics of Figure 1.

Author Response

Thank you for your comments. Figure 1's quality has been improved.
